# The Effect of Nurse Support Programs on Job Satisfaction and Organizational Behaviors among Hospital Nurses: A Meta-Analysis

**DOI:** 10.3390/ijerph192417061

**Published:** 2022-12-19

**Authors:** Se Young Kim, Mi-Kyoung Cho

**Affiliations:** 1Department of Nursing, Changwon National University, Changwon 51140, Republic of Korea; 2Department of Nursing Science, Chungbuk National University, Cheongju 28644, Republic of Korea

**Keywords:** meta-analysis, job satisfaction, organizational commitment, interpersonal relationships, self-efficacy, motivation, burnout, turnover intention

## Abstract

The purpose of this study was to calculate the combined effect size of nurse support programs on job satisfaction among hospital nurses. The PICO framework was used in this methodological study of systematic review and meta-analysis. Using nine electronic databases of four international and five Korean databases and applying the eligibility criteria, articles published from database inception through October 2022 were collected. A total of 24 Korean and international articles were selected following the PRISMA guidelines. The keywords of nurse, mentoring (preceptorship, internship, or residency) program, and job satisfaction were selected based on the PICO. The checklists for RCTs and quasi-experimental studies provided in the Joanna Briggs Institute of Critical Appraisal Tool were used for the quality assessment. The coded data were analyzed using MIX 2.0 statistical software. We analyzed the combined effect sizes, heterogeneity, funnel plot, Egger’s regression test, Begg’s test, subgroup analyses, and univariate meta-regression. The overall effects of the program on job satisfaction and organizational behavior such as organizational commitment, interpersonal relationships, self-efficacy, motivation, burnout, and turnover intention for hospital nurses were statistically significant. The results of this study may explain the effect of the nurse support program on job satisfaction and organizational behaviors for hospital nurses.

## 1. Introduction

In many countries, healthcare providers experience difficulties in recruiting and retaining nurses [1]. Job satisfaction, which is negatively associated with nurses’ turnover intention [2,3], is the most reliable variable for detecting turnover intention [4]. Improving nurses’ job satisfaction is important to address the current domestic and global nurse shortages [5,6]. High turnover has made it essential for nursing managers to adopt strategies that improve nurses’ job satisfaction [7].

Low job satisfaction among nurses is a major issue in healthcare, and various efforts have been made for its improvement. To mitigate this problem, Uys et al. [8] developed a management model, DiMeglio et al. [9] conducted a team-building intervention, and Porter et al. [10] implemented a nurse management partnership. Based on a systematic review, Niskala et al. [11] conducted educational and organizational interventions consisting of training sessions, mentoring programs, interpersonal improvement programs, and evidence-based nursing management practices to improve nurses’ job satisfaction; in particular, they identified the influence of interventions on improving professional self-concept and spiritual intelligence.

In a systematic review, Lin et al. [12] analyzed 11 non-experimental studies and found that residency programs positively affected nurses’ job satisfaction. However, the researchers suggested that a follow-up study should be performed including randomized controlled trials (RCTs) to investigate causality [12]. Niskala et al. [11] performed a meta-analysis of intervention studies on the improvement of job satisfaction using English and Finnish terminologies. In most studies, they found that interventions improved job satisfaction; however, the integration effect was not statistically significant. Of the 20 studies included, 5 were single-arm trials, and only 2 were RCTs, which were insufficient to conclude causality on the effect of interventions. This is because control groups that have different treatment outcomes are essential to enhance the validity of interventions in experimental studies.

Therefore, studies that systematically analyzed the effects of education and training programs for nurses on job satisfaction did not draw consensus on the effectiveness of such programs and included single-group interventions or non-experimental studies in the analysis subjects, limiting the inference of causality [13]. So far, existing meta-analyses only considered articles published in English; thus, their relevance may be overestimated due to the exclusion of articles published in other languages [14,15,16,17,18,19]. Therefore, this study comprehensively named the various education and training, programs, workshops, and management activities developed and conducted for nurses as nurse support programs. Furthermore, it analyzed the results of RCTs and quasi-experimental studies, including dissertations, to calculate the combined effect size of programs to promote hospital nurses’ job satisfaction and provide a basis for developing effective programs that can improve their job satisfaction. 

## 2. Method

### 2.1. Study Design

This is a methodological systematic review and meta-analysis conducted to merge the effect sizes of programs to promote hospital nurses’ job satisfaction according to the Population, Intervention, Comparison, Outcome (PICO) framework. Additionally, it identifies the characteristic factors of programs affecting job satisfaction. 

### 2.2. Eligibility Criteria and Outcome Variables

This study was conducted according to PRISMA [20], and the report was prepared by referring to the PRISMA 2020 checklist (https://prisma-statement.org/PRISMAStatement/Checklist.aspx accessed on 15 December 2022). In line with the purpose of this study, a systematic literature search was conducted based on Population, Intervention, Comparison, Outcome, and Study Design (PICO-SD). The inclusion criteria were as follows: The study population (P) included the general nurse population working in hospitals; the intervention (I) entailed providing support or educational intervention to promote job satisfaction; the control (C) group consisted of hospital nurses who worked in the usual setting or received existing interventions in the hospital aimed at promoting their job satisfaction; and regarding outcomes (O), the primary outcome was job satisfaction, while the secondary outcomes were organizational behaviors such as organizational commitment, interpersonal relationships, self-efficacy, motivation, job stress, burnout, and turnover intention. The first post-intervention value obtained was used for the calculation of the effect size. The study design (SD) involved RCTs and a quasi-experimental study, which included the unpublished manuscript. The exclusion criteria were as follows: the population (P) that included nurses who did not work in a hospital or whose jobs did not include nursing, although they were working in a hospital; the intervention (I) that did not present its effects as a mean, standard deviation, or sample size, such that effect sizes could not be merged; the outcomes (O) that measured program satisfaction without measuring job satisfaction as a variable; and the study design (SD) that used single-group comparative studies from quasi-experimental studies. According to the inclusion and exclusion criteria presented in Table 1, Korean domestic and international electronic databases were searched, and papers published in English or Korean were selected.

### 2.3. Search Strategy

Based on the National Library of Medicine’s (NLM) COre, Standard, Ideal (COSI) model [21], preliminary searches were conducted following the above-listed criteria from 11 July 2022, against e-journals and nine databases, including four international (PubMed, Embase, CINAHL, and the Cochrane Library) and five Korean domestic databases, Research Information Sharing Service (RISS), Korean studies Information Service System (KISS), Kyoboscholar, and DBpia (Database Periodical Information Academic), for which keywords for job satisfaction programs for nurses were used. Keywords selected according to the PICO framework were checked through PubMed’s MeSH database. From 17 September 2022, to 7 October 2022, a full-scale search with keywords such as “nurses”, “job satisfaction”, “mentoring”, “preceptorship”, and “internship and residency” was performed to select articles published online or offline up to 31 October 2022. The search terms were adjusted according to the database (e.g., including programs in intervention in Korean domestic databases) and MeSH terms and text words were used appropriately. The search protocol was registered in the Prospero (registration no. CRD42022360943 available at https://www.crd.york.ac.uk/prospero/#searchadvanced) on 29 September 2022. Additionally, we conducted a comprehensive manual search of systematic reviews or meta-analysis studies on the subject, references of systematic review (SR) studies, references of studies adopted as the subject of analysis, and Google to avoid any potentially missed papers in keyword-driven searches. As a search strategy, we had other researchers verify that the search was conducted using tools such as the peer review of the electronic search strategy checklist, which ensured the validity of the search.

The PRISMA Statement [22] proposed that the search process comprised three steps: identification, screening, and inclusion. After removing redundant papers, the selection and exclusion criteria in Table 1 were applied (Figure 1).

### 2.4. Data Extraction

Throughout the process of data collection and screening, all articles included in the analysis were independently reviewed by two researchers (MKC and SYK). After the two researchers shared the search formulas and independently collected data, they discussed the papers together to identify any discrepancies during each step of the title/abstract/full manuscript review process and then came to a consensus on inclusion and exclusion. The independently collected papers from the search were summarized using Excel for a stepwise method, according to the review process, from which the paper data were extracted for analysis by dividing them by the number or color indicated in the selection and exclusion reasons. First, the title/abstract of the papers were evaluated in the database search, and redundant papers were removed by sorting them by title and author using the filtering function of the Excel program. According to the inclusion and exclusion criteria presented in Table 1, the title/abstract/full manuscript were reviewed step-by-step. Accordingly, papers that did not meet the criteria were excluded, leading to the selection of the final papers. The studies selected for the final analysis were assigned a serial number according to the title and listed in a folder. The extracted characteristics of the studies were author(s), year of publication, the status of publication, number of participants, characteristics of participants, study design, types of programs, program facilitator, program duration, program session, session time, outcome variables, and quality assessment scores. The variables were author(s), year of publication, mean, and standard deviation, which were coded in the order of the number of experimental groups, mean, standard deviation, and the number of control groups. After discussion by the two researchers, only organizational commitment, interpersonal relationships, self-efficacy, motivation, job stress, burnout, and turnover intention were extracted as organizational behaviors.

### 2.5. Quality Assessment

The quality of the included papers was assessed using a checklist of RCTs and quasi-experimental studies included in the Joanna Briggs Institute of Critical Appraisal Tool [23]. A total of 13 items were assessed for RCT quality and 9 for a quasi-experimental study, with scores of 0 (for “no” or “unclear”) and 1 (for “yes”), for a total score of 13 and 9, respectively (https://jbi.global/critical-appraisal-tools accessed on 15 December 2022). Two researchers (MKC and SYK) independently performed the quality assessment of the selected papers using the checklist, before which two pilot tests were performed for each type of study. In the pilot test, there were disagreements on two questions in the checklist items of the RCT, including item numbers 8 (follow-up complete and, if not, adequately described) and 11 (analyzed and reliable method of outcome measures), and three questions in the checklist items of the quasi-experimental study, including item numbers 3 (exposure to similar treatment), 6 (follow-up complete and, if not, adequately described), and 9 (appropriate statistical analysis). Hence, the researchers discussed the content of the evaluated papers together, referring to the checklist manuals, and reached the same evaluation results for the questions.

### 2.6. Data Analysis

The coded data were analyzed using MIX 2.0 Professional software for meta-analysis in Excel, version 2.016 (BiostatXL, Mountain View, CA, USA). To merge effect sizes into continuous variables, the means, standard deviations, and group sample sizes of the first measured posttest values of the experimental and control groups were extracted from each study, and the mean differences were calculated. Cohen’s d tends to overestimate the effect size of the individual study when the sample is small, thus the corrected effect size known as Hedge’s g was presented along with 95% Confidence Intervals (CI) [24,25]. As for the interpretation of the effect size of Hedge’s g, Hedge’s g = 0.15, 0.40, and 0.75 were interpreted as small, medium, and large effects, respectively [26]. The heterogeneity of the study (effect size) was tested by calculating the value of Cochran’s Q, and the inconsistency was examined using Higgins’s I^2^. The significance probability for the Q value was less than 0.05. If I^2^ was more than 50%, it was interpreted as heterogeneity. Higgins’s I^2^ was used in this study because the power of Cochran’s Q is low when the number of studies is small, whereas the power increases when the number of studies is large [27]. In the case of the overall effect and subgroup analyses based on study characteristics, the effect size was calculated as a random effect model [28] to reset the weights, considering the problem of reporting the chosen grouping study, the variation between the participants of the individual studies, and the heterogeneity between the studies [24]. The weights of each effect size were calculated using the inverse of variance [24]. To identify the causes of heterogeneity between studies, subgroup analyses were performed based on the characteristics of the studies: publication status, number of participants, participants’ characteristics, study design, types of programs, program facilitator, program duration, program session, session time, outcome variables, and quality assessment score. As for the meta-regression analysis, univariate meta-regression was performed to test the effect size and relevance by including only one control variable in the analysis, to which all control variables applied in the subgroup analysis were added. The analysis process applied to the meta-regression method was carried out by the fixed-effect model as well as the method of moments, 95% CI, two-sided p-value, and Z-value method of estimating the actual variance. Publication bias refers to the tendency to publish more if the intervention effect is large and statistically significant and is used to test the validity of studies in meta-analyses that integrate and analyze individual studies [29]. Publication bias was visually analyzed using a funnel plot and a trim-and-fill plot and confirmed using Egger’s regression test, Begg’s test, and the trim-and-fill method.

## 3. Result

### 3.1. Data Extraction

The total number of studies found in the database search was 4589 (265 from PubMed, 765 from Embase, 95 from CINAHL, 69 from the Cochrane Library, 2975 from RISS, 24 from KISS, 431 from DBpia, 59 from E-article, and 50 from Kyoboscholar). Of these, the number of studies remaining after removing duplicate literature was 3437, with 1186 redundant papers in each database. After the exclusion of 503 redundant papers and 3 studies for which the original text could not be found when reviewing all the databases together, 2931 studies remained. The final 24 studies were selected after removing 2907 studies by applying the selection and exclusion criteria again. In addition to the systematic search, 70 studies were identified through manual search (36 on websites and 34 in citation search). However, all of these studies were excluded from the analysis due to overlapping and differences in study designs and variables (Figure 1).

### 3.2. Characteristics of the Studies

Table 2 shows the characteristics of the studies analyzed in this study. The papers were published from 2004–2021, and nine papers (37.5%) were published within the last five years (2018–2022). Five studies (20.8%) received funding. The sample sizes were 23–89 participants: 13 studies (54.2%) with more than 50 participants; 16 studies (66.7%) with nurses with less than three years of experience; 3 RCTs (12.5%) in study design with a total of 171 participants; and 21 quasi-experimental studies (87.5%) with 1113 participants. The types of interventions performed included a New Nurse Support Program in 10 (41.7%) studies and a General Nurse Support Program in 14 (58.3%). As for program facilitators, researchers conducted 15 studies (62.5%). The program duration was greater than four weeks in 18 studies (75%) and more than six weeks in 15 studies (62.5%). In total, 20 studies (83.3%) had a total of eight sessions or fewer, and 17 studies (70.8%) had a single session lasting two hours or less. Job satisfaction as the primary outcome variable was assessed in all 24 studies, whereas the organizational behaviors were assessed in 12 studies for organizational commitment, 4 each for interpersonal relationships and motivation, 7 for self-efficacy, 5 each for job stress and burnout, and 9 for turnover intention.

### 3.3. Methodological Quality

The mean quality score for the 3 RCTs was 7.67 (range: 6–10) and the score for the 21 quasi-experimental studies was 8.19 (range: 7–9). The similarity of treatment, participants’ analysis in the groups, the same way of measuring outcomes, and statistical analysis were clearly described in the three RCTs. In all three studies, neither the blinding of participants nor the blinding of those delivering treatment was controlled. As for the quasi-experimental study, four questions, including the clarity of cause-and-outcome effect, comparison of the treated group, same way of measuring outcomes, and appropriate statistical analysis, were clearly explained in all 21 studies, and the similarity of treatment groups was homogeneous in only 14 studies. Based on the evaluation of the study quality, it was judged that the quality of the selected papers was unlikely to change the conclusions of the study (Table 3).

### 3.4. Effects of Nurse Support Programs on Job Satisfaction

For the 24 included studies, the standardized mean difference (Hedge’s g) between the experimental and control groups was calculated using the mean, standard deviation, and sample size, with results presented as 95% CI, weight, and synthesis forest plots (Figure 2). When the effect sizes of the studies were merged, analysis using a random effect model showed a statistically significant increase in job satisfaction (Z = 5.04, *p* < 0.001), and the overall effect size of the program, Hedge’s g = 1.12, was found to be greater than 0.75 when judged by the effect size interpretation presented by Brydges [26]. In the heterogeneity test, there was a variance between studies with Q = 288.97 (Q-*df* = 263.97, *p* < 0.001), and Higgins’s I^2^ was 92.0% in the inconsistency test, confirming the high degree of heterogeneity of the combined effect size. Therefore, subgroup analysis and meta-regression analysis were performed for an exploratory explanation of heterogeneity.

Of the 10 study characteristics, including the publication status, the number of participants, participants’ characteristics, study design, types of programs, program facilitator, program duration, program session, session time, and quality assessment score, “study design” was found to have large effect sizes for both quasi-experimental studies (Hedge’s g = 0.78, 95% CI: 0.49, 1.07) and RCTs (Hedge’s g = 4.63, 95% CI: −0.51, 9.76). However, RCTs were found to be not significant, though the effect size was large. Additionally, effect sizes found in subgroup analyses according to the publication status, number of participants, participants’ characteristics, types of programs, program facilitator, program duration, program session, session time, and quality assessment score were all statistically significant (*p* < 0.001). If the nurse’s work experience was greater than three years and the quality assessment was lower than average, a program of four weeks or less showed a moderate effect (Table 4).

We performed meta-regression analyses to explain the potential effect of study heterogeneity on effect size, depending on study characteristics or differences in study populations. A univariate meta-regression was found to have a statistically significant effect on the applied study design, facilitator of interventions, intervention duration, session, and quality assessment score. RCTs (Z = 4.80, *p* < 0.001) showed higher job satisfaction than quasi-experimental studies. Additionally, the effect size of job satisfaction was more positively affected when the facilitator of intervention was a more qualified researcher for program application or the program had more of a team approach (Z = 2.03, *p* = 0.043); when the program duration was greater than four weeks, rather than below four weeks (Z = 3.99, *p* < 0.001); when the program was composed of more than eight sessions rather than eight or fewer sessions (Z = 3.11, *p* = 0.002); and when the quality evaluation score was higher than average (Z = 4.26, *p* < 0.001) (Table 5).

The basic principle of the sensitivity test is that the dataset is constructed according to the author’s judgment, such as assumptions about quality or study size, and then the effect-on-effect size estimation is examined by checking whether the effect size derived from the changed dataset differs from the original effect size [54]. In this study, the analysis was repeated while sequentially excluding each study from the 24 studies. When the combined effect size and statistical significance of the nurse support programs were examined, Hedge’s g had a large effect size of 0.08-1.12, and the 95% CI (0.49–0.76, 1.11–1.64) did not include zero, all of which were statistically significant (*p* < 0.001). Therefore, the exclusion sensitivity test showed no significant difference in effect size from Hedge’s g = 1.12, which included all 24 studies, and all were statistically significant. Therefore, the meta-analysis of this study was considered robust (Table 6).

### 3.5. Effect of Intervention Programs on Organizational Behaviors

Organizational behaviors of this study included organizational commitment, interpersonal relationships, motivation, self-efficacy, job stress, burnout, and turnover intention, of which all except job stress were statistically significant. After the program, some organizational behaviors significantly increased with a large effect size, as shown by Hedge’s g values, such as 0.94 (95% CI: 0.40, 1.48) for organizational commitment, 1.59 (95% CI: 0.46, 2.71) for interpersonal relationships, and 1.39 (95% CI: 0.44, 2.34) for self-efficacy. Hedge’s g = 0.48 (95% CI: 0.02, 0.93) indicates that the motivation significantly increased with a moderate effect size. The programs significantly decreased burnout and turnover intention, as shown by Hedge’s g = −0.92 (95% CI: −1.35, −0.49) and Hedge’s g = −0.85 (95% CI: −1.24, −0.45), respectively (Table 7).

### 3.6. Publication Bias Analysis and the Overall Risk of Bias

To check for any bias in the study data, we performed a funnel plot analysis to determine whether the papers were symmetrical around the aggregated mean size (Table 8).

According to the result, the individual effect sizes (blue circles) of the 24 papers included in this study were asymmetrical in the funnel plot, skewed to the right side, indicating publication bias to some extent (Figure 3).

The Egger’s regression test for further analysis of publication bias showed that the significance probability of the initial intercept of the regression was statistically significant (*p* < 0.001), and Begg’s test for ranking correlation showed 0.38 for Tau b, 0 for ties, and *p* = 0.010, indicating a publication bias in both results (Table 8). The trim-and-fill method is intended to estimate the number of missing or unreported studies and the corresponding changes in effect estimates, which are used to compare the difference between the original number of papers and the number of corrected papers with the difference in the effect size. Finally, we used the trim-and-fill plot [55] to determine the extent of the impact of publication bias on the study results. The number of papers that needed to be corrected and added to the 24 papers included in this study was identified as 1 (white circle) (Figure 3). While the effect size of 24 papers was 1.12 (95% CI: 0.68, 1.56), that of 25 corrected papers was 0.67 (95% CI: 0.55, 0.79). The effect size of job satisfaction after the programs decreased from a large effect size before calibration to a medium effect size after calibration. However, its statistical significance was confirmed both before and after calibration. Therefore, despite the existence of publication bias in this study, the significance analysis results for the combined effect size of job satisfaction showed no change due to publication bias, and thus the effect size in this meta-analysis study is acceptable.

## 4. Discussion

This study defined various education and training conducted for nurses as “support programs” and examined the effects of these programs on nurses’ job satisfaction using RCTs and quasi-experimental studies reported in publications and dissertations. This study analyzed 24 studies and found that job satisfaction increased significantly in the experimental group with the application of a support program (*p* < 0.001). These results are consistent with Lin et al.’s [12] analysis of nurse residency programs in a meta-analysis of 11 non-experimental studies that found a positive effect of nurse residency programs on job satisfaction for new nurses. It was somewhat similar to Niskala et al.’s [11] meta-analysis of interventions to increase nurses’ job satisfaction, in which most interventions improved job satisfaction, though not in a statistically significant way. Therefore, this study is meaningful because it indicates that support programs for nurses should improve job satisfaction in a statistically significant way.

In this study, the effects of the support programs were analyzed by dividing the participants into new nurses and experienced nurses, and the effect of the programs on improving job satisfaction was significant in both groups. If we look at the content of the 24 support programs analyzed in this study, many studies used cognitive training strategies for nurses, such as self-awareness, self-efficacy, empowerment, meaning, the value of work, potential enhancement, spiritual intelligence, coaching, and stress relief. Hayes’s research [56] found that professional self-concept [57] and positive perceptions of the nursing profession were among the main factors contributing to nurses’ job satisfaction [58,59,60,61]. Moreover, a meta-analysis of interventions to improve nurses’ job satisfaction [11] found that spiritual intelligence training protocols [62] and professional self-concept development programs [63] significantly improved job satisfaction [11]. Furthermore, Judge et al. [64] introduced the concept of core self-assessment mediating between intrinsic job characteristics and job satisfaction, which consists of self-esteem, self-efficacy, control, and emotional stability. Judge et al. [64] found that individuals with positive core self-assessment perceptions were more aware of their intrinsic value. Therefore, support programs that improve nurses’ self-awareness and self-efficacy would increase nurses’ positive perceptions of the value of the nursing profession and ultimately contribute to job satisfaction.

In this study, support programs for the nurse group with less than three years of clinical experience were significantly effective in ensuring job satisfaction. In particular, the support program for new nurses consisted of training to help them adapt to clinical environments, such as residency programs, mentoring, nurse externship programs, and organizational socialization. Lin et al. [12] analyzed the impact of residency programs on new nurses’ job satisfaction and found that positive interpersonal relationships and interactions during residency programs affected new nurses’ satisfaction. Moreover, interaction opportunities with other new nurses provided a sense of belonging and support and also improved job satisfaction and retention levels for new nurses [65,66]. It was also mentioned that engagement with peer nurses should be a key component of programs that improve job satisfaction and retention. Similarly, the Huddling Program developed by Im et al. [67] significantly increased organizational engagement and empowerment of the experimental group by engaging new nurses in peer group activities. However, new nurses participating in the residency program displayed no change in job satisfaction regarding vacation, salary, benefits, scheduling, professional opportunities, work environment, and hospital system [12]. Conversely, Zangaro and Soeken [19] pointed out that autonomy and cooperation at Magnet Hospital were key factors in professional nursing practice, affecting job satisfaction in a meta-analysis, and there were few ways to improve the nurses’ work environment to increase job satisfaction [68,69]. Therefore, an experimental study should be conducted to improve organizational factors, aimed at understanding the effects of changes in staffing, training support, scheduling, head nurse leadership, and human resource management on job satisfaction.

Moreover, it was found that the more qualified researcher or team approach as a facilitator conducting the program, the greater the improvement in nurses’ job satisfaction. Therefore, to effectively achieve the desired outcomes through the program, nursing departments need to employ qualified instructors to approach them as a team when developing and delivering a support program for nurses. Moreover, both a support program lasting more than four weeks and a program consisting of more than eight sessions were found effective in increasing nurses’ job satisfaction. Meanwhile, organizational commitment, interpersonal relationships, and self-efficacy were significantly increased as a result of support programs. Nurses’ job satisfaction and organizational engagement are known to be positively correlated [70]. In particular, the internal factors of job satisfaction, such as technology utilization, job diversity, experience, and service, rather than external factors such as salary, promotion, and work conditions, reinforced their normative organizational engagement [71]. Thus, it was determined that the nurse support program stimulated the internal factors of job satisfaction, leading to an increase in nurses’ job satisfaction and related organizational engagement.

Conversely, Zangaro and Soeken [19] pointed out that autonomy and cooperation at Magnet Hospital were principal factors in the professional nursing practice, affecting job satisfaction in a meta-analysis, and there were few ways to improve the nurses’ work environment to increase job satisfaction [68,69]. These results were seemingly attributable to the provision of support programs primarily to a single group by nursing organizations. Moreover, the topics and content of the support programs analyzed in this study varied substantially, so it was difficult to determine which education and training program affected nurses’ job satisfaction. Therefore, we believe that various nursing organizations and schools should collaborate to develop standardized nurse support programs and apply them sequentially in various institutions in the future, allowing the programs’ effects to be objectively compared and analyzed. Additionally, this study was limited to English and Korean literature. Therefore, systematic reviews and meta-analyses of papers written in various languages are needed in the future.

## 5. Conclusions

Support programs for nurses are effective in improving the job satisfaction of new and experienced nurses. Support programs that provided training to new nurses to adapt to clinical practice and cognitive strategies to improve their occupational awareness and self-assessment were more effective in improving their job satisfaction when delivered by a qualified facilitator or team over a minimum of eight sessions spread out over four weeks. The results of this study can be used as evidence for the development of support programs for nurses in the future, suggesting the need for organizational changes and interventions that affect nurses’ job satisfaction. In conclusion, nursing organizations should develop and continuously provide customized support programs that reflect the needs of each group of new and experienced nurses to improve nurses’ job satisfaction and effectively retain them.

## Figures and Tables

**Figure 1 ijerph-19-17061-f001:**
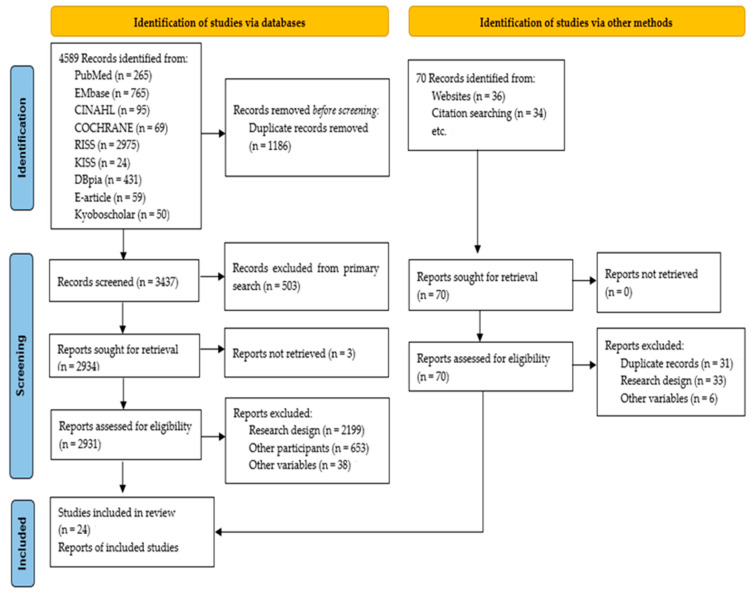
PRISMA flow diagram.

**Figure 2 ijerph-19-17061-f002:**
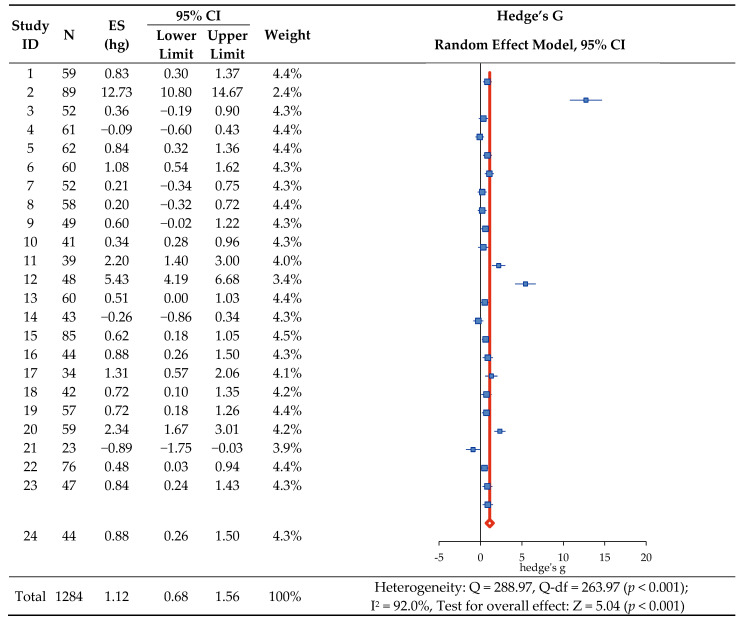
The effect of nurse support program on job satisfaction. Notes. ES: Effect size; CI: Confidence interval.

**Figure 3 ijerph-19-17061-f003:**
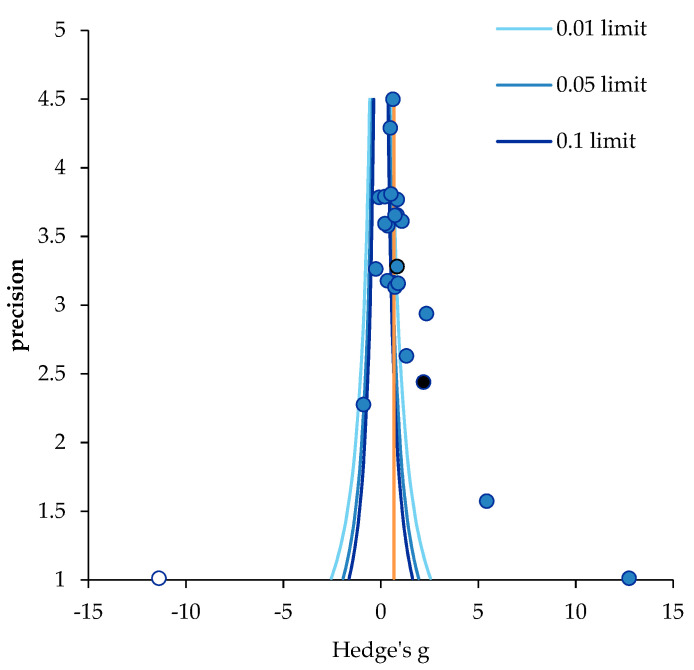
Funnel plot of nursing support program on job satisfaction. Notes. Precision = 1/standard error; 0.05; limit line = 95% confidence limit.

**Table 1 ijerph-19-17061-t001:** Study eligibility criteria.

PICO-SD	Inclusion Criteria	Exclusion Criteria
Participants	Hospital nurse General nurse	Not a hospital nurse or a nurse manager
Intervention	Nurse support program Hospital setting Studies published through 31 October 2022 Studies published in English or Korean Including unpublished research	Non-hospital setting Studies not published until 31 October 2022
Control	Usual or comparative experiments	
Outcomes	Primary outcome: Job satisfaction	Did not measure job satisfaction as an outcome variable
	Secondary outcome: Organizational behaviors such as organizational commitment, interpersonal relationships, self-efficacy, motivation, job stress, burnout, and turnover intention After the program, the first post-test value was used as the post-test value Studies report means, standard deviation, and concrete sample sizes	Studies report visible graphs or only p-value Studies in which job satisfaction was measured but the effect size could not be calculated
Study design	Randomized controlled trial (RCT) Quasi-experimental design	Not quasi-experimental studies or RCT Single-group comparative study
		Survey

**Table 2 ijerph-19-17061-t002:** Descriptive summary of the included studies.

Study ID	Author	Year	Publication	Participants		Study Design	Program Type	Program Facilitator	Program Duration (Week)	Program Session (Frequency)	Session Time (Hour)	Outcome Variable	Quality Score
1	Chen et al. [30]	2010	Yes	59 psychiatric hospital nurses (E: 26, C: 33)	Mean work experience: 12 years	Quasi-E	Potentiality education program	Doctoral and master students, manager	8	4	3.5	Job satisfaction and potentiality	8
2	Sampson et al. [31]	2019	Yes	89 new licensed RNs (E: 47, C: 42)	New nurses	Cluster RCT	Residency program	Researcher	8	8	0.75	Job satisfaction, stress, depressive symptoms, anxiety, and healthy lifestyle beliefs and behaviors	10
3	Cantrell et al. [32]	2006	Yes	52 general nurses (E: 26, C: 26)	Work experience: 6 months–3 years	Quasi-E (paired matching)	Nurse externship program	1:1 preceptor	10	NA	NA	Job satisfaction, professionalism, role socialization, and sense of belonging	8
4	Park [33]	2004	Yes	61 general nurses (E: 23, C: 38)	Work experience: diverse	Quasi-E	Value clarification training	Researcher	8	8	1.5–2	Job satisfaction, job motivation, and professional self-concept	8
5	Han [34]	2019	No	62 general nurses (E: 31, C: 31)	Work experience: diverse	Quasi-E	Work meaning program based on logotherapy	Researcher	8	4	1–1.5	Job satisfaction, meaning in life, work meaning, wellness, job commitment, and intrinsic motivation	8
6	Baek [35]	2016	Yes	60 general nurses (E: 30, C: 30)	Work experience: 3–10 years	Quasi-E	Coaching program	Researcher: coaching qualifications	8	8	1–1.5	Job satisfaction, emotional intelligence, coaching skill, and self-efficacy	8
7	Jang [36]	2019	No	52 elderly nursing hospital nurses (E: 27, C: 25)	More than 3 months of work experience	Quasi-E	Intentional nursing rounds protocol	Researcher	2	1, 10/day protocol application	1.5	Job satisfaction, communication competence, clinical work competence, and compassionate competence	7
8	Bae [37]	2021	No	58 general nurses (E: 27, C: 31)	3 shifts, 2–6 years nurses	Quasi-E	Meaning-centered job identity program	Researcher	8	8	1.5	Job satisfaction, job identity, the meaning of work, internal motivation, and resilience	9
9	Choi et al. [38]	2016	Yes	49 general nurses (E: 34, C: 15)	2-year nurses	Quasi-E	Empowerment program	Researcher	2 days	7	1–3	Job satisfaction, turnover intention, organizational commitment, self-efficacy, and burnout	8
10	Cho [39]	2016	No	41 general nurses (E: 20, C: 21)	3 shifts within 1-year nurses	Quasi-E	Competence Enhancement Program	Researcher	4	7	1–1.5	Job satisfaction, nursing competencies, job performance, interpersonal relationship competence, self-efficacy, self-reflection, and turnover intention	9
11	Lee [40]	2017	No	39 general nurses (E: 20, C: 19)	3 shifts within 1-year nurses	Quasi-E	Case-Based Nursing Organizational Socialization Program	Researcher	6	6	2	Job satisfaction, organizational socialization, organizational commitment, and turnover intention	9
12	Jeong [41]	2015	No	48 general nurses (E: 24, C: 24)	Nurses working within 8 months	Quasi-E	Self-coaching program	Researcher: coaching qualifications	6	6	4	Job satisfaction, coaching behavior (interpersonal ability), organizational loyalty, self-efficacy, and program satisfaction	9
13	Kim [42]	2015	No	60 general nurses (E: 30, C: 30)	3 shifts within 1 year of nurses	Quasi-E	Clinical adaptation promotion program	Researcher	3	3	2	Job satisfaction, nursing performance, self-efficacy, interpersonal relationships, professional self-concept, organizational commitment, and burnout	8
14	Cho [43]	2020	No	43 general nurses (E: 22, C: 21)	3 shifts within 1 year of nurses	Quasi-E	Reduction program in transition shock	Researcher	4	8	1	Job satisfaction, transition shock (stress), turnover intention, organizational commitment, and critical thinking	8
15	Seo et al. [44]	2014	Yes	85 general nurses (E: 41, C: 44)	Nurses working less than 5 years	Quasi-E	Spirituality promotion program	Researcher	8	8	1.5	Job satisfaction, spirituality, perceived stress, positive and negative affect, empathy, and leadership practice	8
16	Lee [45]	2018	No	44 general nurses (E: 21, C: 23)	Nurses working for more than 6 months	Quasi-E	Empowerment promotion program	Researcher	24	8	2	Job satisfaction, self-efficacy, motivation, organizational commitment, and burnout	8
17	Moon et al. [46]	2021	Yes	34 general nurses (E: 17, C: 17)	ER nurses working for more than 6 months	Quasi-E	Positive emotions reinforcement program	Researcher, professional instructor, and psychiatrist	24	5	2	Job satisfaction, positive psychological capital, compassion satisfaction, and compassion fatigue (stress and burnout)	9
18	Lee et al. [47]	2010	Yes	42 general nurses (E: 20, C: 22)	Nurses working for 2–3 years	Quasi-E	Mentoring program	Researcher and mentor	24	10	NA	Job satisfaction, organizational commitment, empowerment, career commitment, and turnover intention	8
19	Yoo [48]	2020	Yes	57 general nurses (E: 29, C: 28)	3 shifts, clinical nurses less than 10 years	Quasi-E	Person-centered nursing educational program	Researcher	24	6	1	Job satisfaction, self-awareness, interpersonal relationship competency, self-esteem, and co-worker support	8
20	Kim et al. [49]	2012	Yes	59 ICU nurses (E: 29, C: 30)	3 shifts, ICU nurses working less than 3 years	RCT	Self-directed critical care nursing e-learning program	Researcher	24	18	0.5	Job satisfaction, knowledge, and the performance of critical care nursing	7
21	Boo [50]	2006	No	23 new nurses (E: 12, C: 11)	New nurses working within 12 months	RCT	Self-efficacy promoting program	Researcher	8	4	NA	Job satisfaction, self-efficacy, organizational commitment, task adaptation	6
22	Choi et al. [51]	2014	Yes	76 general nurses (E: 40, C: 36)	New nurses working within 12 months	Quasi-E	Organizational socialization education program	Researcher and mentor	2 days	3	5–6	Job satisfaction, organizational commitment, and turnover intention	8
23	Kim [52]	2018	Yes	47 general nurses (E: 23, C: 24)	Nurses working for 0.5–3 years	Quasi-E	Group rational emotive behavior therapy	Researcher: REBT qualification	8	8	3	Job satisfaction, job stress, burnout, organizational commitment, and turnover intention	7
24	Lee [53]	2011	No	44 general nurses (E: 22, C: 22)	Nurses working for more than 3 years	Quasi-E	Followership program	Researcher: followership qualifications, professional instructor, and 3 research assistants	6	12	1.5	Job satisfaction, organizational commitment, turnover intention, followership, and program satisfaction	9

Notes. E: experimental group; C: control group; Quasi-E: quasi-experimental study; RCT: randomized controlled trials; NA: not applicable; REBT: rational emotive behavior therapy.

**Table 3 ijerph-19-17061-t003:** Quality assessment of the included studies.

Joanna Briggs Institute of Critical Appraisal Tools Checklist for Randomized Controlled Trials														Total Score
Study ID	1	2	3	4	5	6	7	8	9	10	11	12	13	
2	0	1	1	0	0	1	1	1	1	1	1	1	1	10
20	1	0	1	0	0	0	1	0	1	1	1	1	0	7
21	0	0	0	0	0	0	1	1	1	1	0	1	1	6
Total	1	1	2	0	0	1	3	2	3	3	2	3	2	7.67
**Joanna Briggs Institute of Critical Appraisal Tools Checklist for Quasi-Experimental Studies**														
**Study ID**	**1**	**2**	**3**	**4**	**5**	**6**	**7**	**8**	**9**	**Total Score**				
1	1	0	1	1	1	1	1	1	1	8				
3	1	1	1	1	0	1	1	1	1	8				
4	1	0	1	1	1	1	1	1	1	8				
5	1	0	1	1	1	1	1	1	1	8				
6	1	1	1	1	1	1	1	0	1	8				
7	1	1	0	1	1	0	1	1	1	7				
8	1	1	1	1	1	1	1	1	1	9				
9	1	0	1	1	1	1	1	1	1	8				
10	1	1	1	1	1	1	1	1	1	9				
11	1	1	1	1	1	1	1	1	1	9				
12	1	1	1	1	1	1	1	1	1	9				
13	1	0	1	1	1	1	1	1	1	8				
14	1	1	1	1	1	0	1	1	1	8				
15	1	0	1	1	1	1	1	1	1	8				
16	1	1	0	1	1	1	1	1	1	8				
17	1	1	1	1	1	1	1	1	1	9				
18	1	0	1	1	1	1	1	1	1	8				
19	1	1	1	1	1	0	1	1	1	8				
22	1	1	0	1	1	1	1	1	1	8				
23	1	1	0	1	1	0	1	1	1	7				
24	1	1	1	1	1	1	1	1	1	9				
Total	21	14	17	21	20	17	21	20	21	8.19				

**Table 4 ijerph-19-17061-t004:** Subgroup analysis regarding job satisfaction by study characteristics.

Characteristics	Subgroup	K	Study ID	N	Overall ES	95% CI		Z (*p*)	I^2^ (%)
						Lower Limit	Upper Limit		
Publication	No	11	5, 7, 8, 10, 11, 12, 13, 14, 16, 21, and 24	514	0.85	0.24	1.45	2.76 (0.006)	90.0
	Yes	13	1, 2, 3, 4, 6, 9, 15, 17, 18, 19, 20, 22, and 23	770	1.38	0.74	2.02	4.23 (<0.001)	93.5
Participants (person)	<50	11	9, 10, 11, 12, 14, 16, 17, 18, 21, 23, and 24	454	1.02	0.38	1.66	3.10 (0.002)	89.6
	≥50	13	1, 2, 3, 4, 5, 6, 7, 8, 13, 15, 19, 20, and 22	830	1.22	0.61	1.83	3.92 (<0.001)	93.8
Career (year)	≤3	16	2, 3, 7, 9, 10, 11, 12, 13, 14, 16, 17, 18, 20, 21, 22, and 23	798	1.49	0.79	2.18	4.19 (<0.001)	94.5
	>3	8	1, 4, 5, 6, 8, 15, 19, and 24	486	0.63	0.36	0.89	4.60 (<0.001)	51.8
Study design	Quasi-E	21	1, 3, 4, 5, 6, 7, 8, 9, 10, 11, 12, 13, 14, 15, 16, 17, 18, 19, 22, 23, and 24	1113	0.78	0.49	1.07	5.32 (<0.001)	80.7
	RCT	3	2, 20, and 21	171	4.63	−0.51	9.76	1.77 (0.078)	98.8
Interventions	New nurse support program	10	2, 7, 10, 11, 12, 13, 14, 17, 21, and 22	505	1.95	0.82	3.09	3.37 (0.001)	96.4
	General nurse support program	14	1, 3, 4, 5, 6, 8, 9, 15, 16, 18, 19, 20, 23, and 24	779	0.75	0.49	1.02	5.60 (<0.001)	68.0
Facilitator of intervention	Researcher only	15	2, 4, 5, 7, 8, 9, 10, 11, 13, 14, 15, 16, 19, 20, and 21	822	1.11	0.48	1.73	3.48 (0.001)	93.8
	Qualified or team approach	9	1, 3, 6, 12, 17, 18, 22, 23, and 24	462	1.17	0.62	1.73	4.14 (<0.001)	86.6
Duration of program (week)	≤4	6	7, 9, 10, 13, 14, and 22	321	0.33	0.09	0.57	2.74 (0.006)	10.7
	>4	18	1, 2, 3, 4, 5, 6, 8, 11, 12, 15, 16, 17, 18, 19, 20, 21, 23, and 24	963	1.46	0.88	2.04	4.91 (<0.001)	93.5
Duration of program (week)	≤6	9	7, 9, 10, 11, 12, 13, 14, 22, and 24	452	1.03	0.36	1.70	3.01 (0.003)	90.5
	>6	15	1, 2, 3, 4, 5, 6, 8, 15, 16, 17, 18, 19, 20, 21, and 23	832	1.19	0.61	1.78	3.98 (<0.001)	93.1
Sessions (frequency) *	≤8	20	1, 2, 4, 5, 6, 7, 8, 9, 10, 11, 12, 13, 14, 15, 16, 17, 19, 21, 22, and 23	1087	1.15	0.65	1.65	4.49 (<0.001)	92.8
	>8	3	18, 20, and 24	145	1.31	0.32	2.29	2.61 (0.009)	86.0
Operation time of sessions (hour) *	≤2	17	2, 4, 5, 6, 7, 8, 9, 10, 11, 13, 14, 15, 16, 17, 19, 20, and 24	937	1.18	0.65	1.72	4.35 (<0.001)	92.6
	>2	4	1, 12, 22, and 23	230	1.75	0.43	3.06	2.60 (0.009)	94.5
Quality assessment score	Below the mean	17	1, 3, 4, 5, 6, 7, 9, 13, 14, 15, 16, 18, 19, 20, 21, 22, and 23	931	0.59	0.32	0.85	4.32 (<0.001)	74.1
	Above the mean	7	2, 8, 10, 11, 12, 17, and 24	353	3.08	1.39	4.77	3.57 (<0.001)	97.1

Notes. * Missing value; K: number of analysis set; N: number of participants; ES: effect size; CI: confidence interval; I^2^: inconsistency; IRB: institutional review board; Quasi-E: quasi-experimental study; RCT: randomized controlled trials.

**Table 5 ijerph-19-17061-t005:** Meta-regression analysis evaluating job satisfaction.

Covariate (Ref.)	Estimate	SE	Z	*p*
Publication (ref. = No)	0.22	0.12	1.77	0.077
Participants (ref. <50 people)	−0.09	0.13	−0.69	0.488
Career (ref. ≤3 years)	−0.18	0.12	−1.42	0.155
Study design (ref. = Quasi-E)	1.28	0.27	4.80	<0.001
Interventions (ref. = New nurse)	−0.01	0.13	−0.04	0.967
Facilitator of intervention (ref. = researcher)	0.26	0.13	2.03	0.043
Program duration (ref. ≤4 weeks)	0.54	0.14	3.99	<0.001
Program duration (ref. ≤6 weeks)	0.12	0.13	0.94	0.349
Sessions (ref. ≤8)	0.62	0.20	3.11	0.002
Operation time of sessions (ref. ≤2 h)	0.27	0.16	1.65	0.099
Quality assessment score (ref. = Below the mean)	0.67	0.16	4.26	<0.001

Notes. Ref.: reference; SE: standard error; Quasi-E; quasi-experimental study

**Table 6 ijerph-19-17061-t006:** Exclusion sensitivity test of the nurse support program.

Study ID	K	Hedge’s G	95% CI		Z	*p*
			Lower Limit	Upper Limit		
1	23	1.14	0.68	1.60	4.89	<0.001
2	23	0.80	0.49	1.11	5.05	<0.001
3	23	1.16	0.71	1.62	5.00	<0.001
4	23	1.18	0.73	1.63	5.13	<0.001
5	23	1.14	0.68	1.60	4.88	<0.001
6	23	1.13	0.67	1.59	4.86	<0.001
7	23	1.17	0.71	1.62	5.04	<0.001
8	23	1.17	0.71	1.63	5.03	<0.001
9	23	1.15	0.70	1.61	4.96	<0.001
10	23	1.16	0.71	1.62	5.02	<0.001
11	23	1.07	0.63	1.51	4.77	<0.001
12	23	0.94	0.54	1.35	4.60	<0.001
13	23	1.16	0.70	1.62	4.95	<0.001
14	23	1.18	0.74	1.63	5.19	<0.001
15	23	1.16	0.69	1.62	4.89	<0.001
16	23	1.14	0.68	1.59	4.91	<0.001
17	23	1.12	0.67	1.57	4.86	<0.001
18	23	1.15	0.69	1.60	4.94	<0.001
19	23	1.15	0.69	1.61	4.91	<0.001
20	23	1.06	0.62	1.49	4.77	<0.001
21	23	1.20	0.76	1.64	5.35	<0.001
22	23	1.16	0.70	1.62	4.93	<0.001
23	23	1.14	0.69	1.60	4.91	<0.001
24	23	1.14	0.68	1.59	4.90	<0.001

Notes. K: number of analysis set; CI: confidence interval

**Table 7 ijerph-19-17061-t007:** The effect of nurse support program on organizational behaviors.

Variables	K (Study ID)	N	Hedge’s G	95% CI		Z (*p*)	I^2^ (%)
				Lower Limit	Upper Limit		
Organizational commitment	12 (5, 9, 11, 12, 13, 14, 16, 18, 21, 22, 23, and 24)	577	0.94	0.40	1.48	3.42 (0.001)	88.6
Interpersonal relationships	4 (10, 12, 13, and 19)	206	1.59	0.46	2.71	2.75 (0.006)	91.8
Self-efficacy	7 (6, 9, 10, 12, 13, 16, and 21)	325	1.39	0.44	2.34	2.86 (0.004)	92.6
Motivation	4 (4, 5, 8, and 16)	225	0.48	0.02	0.93	2.06 (0.040)	64.2
Job stress	5 (2, 14, 15, 17, and 23)	298	0.39	−1.51	2.30	0.41 (0.685)	97.8
Burnout	5 (9, 13, 16, 17, and 23)	234	−0.92	−1.35	−0.49	−4.23 (<0.001)	57.6
Turnover intention	9 (9, 10, 11, 13, 14, 18, 22, 23, and 24)	441	−0.85	−1.24	−0.45	−4.20 (<0.001)	73.9

Notes. K: number of analysis sets; N: number of participants; CI: confidence interval; I^2^: heterogeneity

**Table 8 ijerph-19-17061-t008:** Publication bias test of nurse support program on job satisfaction.

Publication Bias Test		Coefficient	SE	95% CI		Z	*p*
				Lower Limit	Upper Limit		
Egger’s regression test	Intercept	10.85	2.15	6.63	15.07	5.04	<0.001
	Slope	−2.47	0.65	−3.74	−1.19	−3.79	<0.001
		tau b		ties		Z	*p*
Begg’s test	Standard	0.38		0		2.58	0.010
	Corrected	0.37		0		2.55	0.011
Trim-and-fill method	Hedge’s g	95% CI					
		Lower limit				Upper limit	
Original	1.12	0.68				1.56	
Corrected	0.67	0.55				0.79	

Notes. Egger’s regression test for zero intercepts. Begg’s test for rank correlation. SE: standard error; CI: Confidence interval

## Data Availability

Not applicable.

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
