# Peer review of "The Effect of Nurse Support Programs on Job Satisfaction and Organizational Behaviors among Hospital Nurses: A Meta-Analysis"

_ijerph, 2022, doi:10.3390/ijerph192417061_

Round 1

Reviewer 1 Report

The topic is relevant, the methodological framework is absolutely appropriate and thorough, the discussion of the results is pertinent.

Before congratulating the authors, I would recommend improving the fluidity of the abstract by removing the parentheses with the results, and editing the English as the rendering of the sentences is sometimes difficult to understand.

Author Response

We appreciate the time and effort that you and the reviewers have put into the valuable feedback and insightful comments provided on this manuscript. We have carefully considered each comment and made changes to the manuscript, as required.

We have marked the revisions made to the manuscript in red font.

Reviewer 2 Report

The authors did a big job trying to answer the question how support programs are effective for nurses' job satisfaction. I suggest to rethink the title and the main information of the manuscript. It seems to me that the title should include not only job satisfaction, but also secondary outcomes which were analyzed in the study. Maybe these secondary outcomes could be called organizational behaviour?

The authors should clearly answer to the question do support program and training program are synonyms in this study. The authors use both these concepts in the manuscript and it is not understandable how them interpret. If concepts support and training are synonyms in this study maybe not both but one concept could be used in the manuscript.

I believe that the study is meaningful step trying to answer question what training programs are effective for nurses' job satisfaction, but I miss the disclosure of the problem, the description of the main question of the study and the justification of the importance of the research in the introduction. Also I do not understand what criteria were used deciding to analyze some training programs in the introduction and ignore other.

The results of support programs are analyzed in Discussion, but for understanding what these support programs are the summary of the programs content, the main features of their intervention are needed.

Author Response

(The authors gave the same response as above.)

Reviewer 3 Report

This article is very useful. It provides strong evidence for actions that organizations can take to support nurses.  I am not a statistician, so I am assuming that the statistics presented in the article are correct.  I am assuming this because the whole approach is well thought out and logical, and the discussion and recommendations are a logical interpretation of the results.

Author Response

We appreciate the time and effort that you and the reviewers have put into the valuable feedback and insightful comments provided on this manuscript.